# Nationwide Trends and Projections of Early Onset Gastrointestinal Cancers in China

**DOI:** 10.3390/cancers17182954

**Published:** 2025-09-09

**Authors:** Tianyu Li, Chen Lin, Weibin Wang

**Affiliations:** Department of General Surgery, Peking Union Medical College Hospital, Chinese Academy of Medical Sciences and Peking Union Medical College, Beijing 100730, China; litianyu@student.pumc.edu.cn

**Keywords:** gastrointestinal cancer, early-onset, global burden of disease, China, projections

## Abstract

Early-onset gastrointestinal cancers have emerged as a significant health challenge in China. This study analyzed national data for individuals aged 15–49 years from 1990 to 2021. We found substantial declines in stomach, esophageal, and liver cancers, contrasted with rising trends in colorectal, biliary tract, and pancreatic cancers. Projections through 2040 indicate further increases in colorectal and biliary cancers, with the age-standardized incidence rate of pancreatic cancer expected to peak around 2030. These divergent trajectories highlight the need to adapt prevention, screening, and early detection strategies to early-onset populations. By characterizing site-specific patterns and forecasting future burdens, our findings provide an evidence base to guide public health policy, optimize clinical management, and improve cancer control among individuals affected at younger ages.

## 1. Introduction

Gastrointestinal cancers represent a major global health burden, accounting for substantial morbidity and mortality worldwide [1]. Although these malignancies have traditionally been considered diseases of older adults, growing evidence indicates that early-onset gastrointestinal cancers, defined as those diagnosed between ages 15 to 49 [2,3], are increasingly prevalent and represent a distinct clinical and epidemiological entity [4]. This trend has significant implications for public health planning, clinical practice, and resource allocation.

China, with its large population and shifting epidemiological landscape, provides a unique setting to examine the changing burden of gastrointestinal cancers [5]. Although gastrointestinal cancers have been widely studied, research specifically targeting early-onset cases remains limited. Understanding temporal and demographic patterns among younger cohorts is critical, given their different risk factor profiles, tumor biology, and often poorer prognosis [6].

In this study, we evaluated nationwide trends in the incidence and mortality of early-onset gastrointestinal cancers from 1990 to 2021, including cancers of the esophagus, stomach, liver, colorectum, gallbladder and biliary tract, and pancreas. We also explored age- and sex-specific differences and subtype distributions and applied Bayesian age–period–cohort modeling to project future trends through 2040. The BAPC model is particularly suitable for forecasting cancer burden because it simultaneously captures the effects of age, calendar period, and birth cohort, providing a comprehensive description of long-term temporal dynamics across populations. In addition, as a Bayesian framework, the BAPC model naturally generates uncertainty intervals, which enhance the interpretability of the results and strengthen their value for policy-making.

Our aim is to identify high-risk subgroups and cancer types that warrant targeted interventions, providing an evidence-based foundation to inform screening strategies, preventive measures, and clinical management of early-onset gastrointestinal malignancies in China.

## 2. Method

### 2.1. Overview and Data Collection

We leveraged the Global Burden of Disease (GBD) 2021 study as our primary data source, which synthesizes information from national cancer registries, vital registration systems, household surveys, and hospital records under rigorous quality control by the Institute for Health Metrics and Evaluation (IHME) (Washington, DC, USA) [7]. The GBD 2021 study evaluated the incidence and mortality rates, along with uncertainty intervals, for 371 diseases and injuries across 204 countries, territories, and 811 subnational locations from 1990 to 2021. We accessed incidence rates, mortality rates, and relevant annual case counts for six gastrointestinal cancer sites (esophagus, stomach, liver, colon and rectum, gallbladder and biliary tract, and pancreas) in individuals aged 15–49 years for China from 1990 through 2021 via the GBD Results Tool (https://vizhub.healthdata.org/gbd-results/, accessed on 10 July 2025.) [8]. All the extracted data and subsequent analyses were restricted to individuals aged 15–49 years, in keeping with the study definition of early-onset gastrointestinal cancers, and this age range was applied consistently throughout the modeling and projections.

### 2.2. Temporal Trend Analysis and EAPC Calculation

We evaluated temporal trends by fitting an ordinary least-squares regression of the natural logarithm of each age-standardized rate against calendar year: ln(rate_t) = α + β·t + ε_t, where t is the year and β is the estimated slope. The slope β represents the average annual change on the log scale and was converted to the estimated annual percent change (EAPC) using the formula EAPC = 100 × (exp(β) − 1). A 95% confidence interval for the EAPC was derived from the standard error of β, and a two-sided *p* < 0.05 was considered to indicate a statistically significant upward or downward trend.

### 2.3. AAPC Calculation

When fitting a regression model, in addition to potential linear terms, one or more covariates exhibit piecewise (i.e., broken-line or segmentwise linear) or stepwise (i.e., segmentwise constant) effects. To summarize the fitted piecewise linear relationships, Clegg et al. [9] proposed the Average Annual Percent Change (AAPC), which is the sum of slopes weighted by the corresponding covariate subintervals. The detailed calculation method for AAPC was shown at https://surveillance.cancer.gov/help/joinpoint/setting-parameters/method-and-parameters-tab/apc-aapc-tau-confidence-intervals/average-annual-percent-change-aapc (accessed on 4 September 2025).

### 2.4. Age-Adjusted Rates Calculation

Age-adjusted rates were calculated using the direct standardization method to account for differences in age distribution across populations and over time (https://seer.cancer.gov/seerstat/tutorials/aarates/step3.html, accessed on 10 July 2025). Specifically, age-specific incidence or mortality rates were first obtained for each five-year age group (e.g., 15–19, 20–24, …, 45–49 years). These rates were then weighted using the corresponding age-group proportions from the Global Burden of Disease (GBD) world standard population. The weighted rates were summed and divided by the total standard population, and the final value was multiplied by 100,000 to express the result per 100,000 population. The formula is as follows: age-adjusted rate = (Σ (rᵢ × wᵢ)/Σ wᵢ) × 100,000, where rᵢ is the observed rate in age group i, and wᵢ is the weight of the standard population in that age group. This approach ensures that temporal and regional comparisons are not confounded by differences in age structure.

### 2.5. Bayesian Age–Period–Cohort Projection Model

Future incidence and mortality trends were projected using a Bayesian age–period–cohort (BAPC) model with a Poisson likelihood (log link) and population offset. Random walk priors were assigned to temporal effects: second-order random walk (RW2) for age and cohort and first-order random walk (RW1) for period, with an additional IID component to account for over-dispersion. Precision parameters followed log-Gamma (1, 5 × 10^−5^) priors for age, period, and cohort, and log-Gamma (1, 5 × 10^−3^) for the over-dispersion term, while the intercept was assigned a vague Gaussian prior [10]. Model fitting was performed using integrated nested Laplace approximation (INLA). Posterior predictive checks, including coverage and residual analyses, were used to evaluate model adequacy. Age-standardization was conducted with WHO 2000–2025 standard population weights. Posterior summaries of hyperparameters are provided in the Appendix A.

### 2.6. Case Definitions

Cancer cases and deaths were classified according to the *International Classification of Diseases, 10th Revision* (ICD-10) as follows: esophageal cancer (C15), stomach cancer (C16), liver cancer (C22), colon and rectal cancer (C18–C20), gallbladder and biliary tract cancer (C23–C24), and pancreatic cancer (C25).

### 2.7. Statistical Analysis

We assessed the burden of gastrointestinal cancers using incidence and mortality, along with their age-standardized rates per 100,000 population. All the estimates were accompanied by 95% uncertainty intervals (UIs), derived using the standardized analytical framework of the GBD 2021 study (https://ghdx.healthdata.org/gbd-2021/code, accessed on 10 July 2025). For future projections from 2022 to 2040, Bayesian age–period–cohort (BAPC) models were employed, offering a comprehensive framework for forecasting based on full Bayesian inference using integrated nested Laplace approximations (INLAs) [11]. In addition to the BAPC model, we applied the Autoregressive Integrated Moving Average (ARIMA) model as a supplementary approach to validate our projections. All the statistical analyses and visualizations were conducted using the R software package (version 4.2.3) and JD_GBDR (V2.5, Jingding Medical Technology Co., Ltd., Xiamen, China). A two-sided *p* value < 0.05 was considered statistically significant.

## 3. Result

### 3.1. National Trends in Early-Onset Gastrointestinal Cancer Incidence and Mortality in China, 1990–2021

Between 1990 and 2021 in China, age-standardized incidence rates (ASIRs) for esophageal, stomach, and liver cancers declined markedly, even as absolute case counts followed divergent paths (Table 1; Figure 1A). Esophageal cancer ASIR fell from 4.86 to 2.18 per 100,000 (estimated annual percent change [EAPC] −3.23%, 95% CI −3.47 to −2.99). Stomach cancer ASIR decreased from 12.87 to 7.05 per 100,000 (EAPC −2.10%, 95% CI −2.21 to −2.00), with case numbers falling from about 70,250 to 58,120. Although liver cancer case counts increased from approximately 30,185 to 38,508, its ASIR declined modestly from 5.47 to 4.63 per 100,000 (EAPC −0.84%, 95% CI −1.14 to −0.53). In contrast, colorectal and gallbladder/biliary tract cancers emerged as growing public health concerns: colorectal cancer ASIR rose from 6.19 to 9.67 per 100,000 (EAPC 1.44%, 95% CI 1.16 to 1.73) alongside an increase in cases from roughly 35,027 to 78,692, while gallbladder and biliary tract cancer ASIR climbed from 0.38 to 0.47 per 100,000 (EAPC 0.92%, 95% CI 0.73 to 1.11) as cases grew from about 2034 to 3893. Pancreatic cancer ASIR experienced only a slight uptick from 1.15 to 1.23 per 100,000 (EAPC 0.17%, 95% CI 0.04 to 0.30), with absolute case counts increased from approximately 6240 to 10,301.

To more accurately capture the non-linear characteristics of the incidence trends of early-onset gastrointestinal cancers, the Average Annual Percent Change (AAPC) analysis was performed (Table 2). For esophageal cancer, the incidence AAPC was −0.996 (95% CI: −1.213 ~ −0.778, *p* < 0.001), and for stomach cancer, it was −0.579 (95% CI: −0.717 ~ −0.440, *p* < 0.001), indicating a continuous downward trend in their incidences. The incidence AAPC of liver cancer was 0.721 (95% CI: 0.489 ~ 0.954, *p* < 0.001), suggesting an upward trend. Notably, colorectal cancer had a high incidence AAPC of 2.770 (95% CI: 2.650 ~ 2.890, *p* < 0.001). Gallbladder and biliary tract cancer showed an incidence AAPC of 2.163 (95% CI: 1.990 ~ 2.335, *p* < 0.001), and pancreatic cancer had an incidence AAPC of 1.638 (95% CI: 1.545 ~ 1.730, *p* < 0.001). These results clearly demonstrated that the incidences of these cancer types were on a significant upward trend. The segment-specific AAPC analysis of incidence and death rate of early-onset gastrointestinal cancers at different year intervals is shown in Appendix A.

Over the same period, mortality patterns largely mirrored incidence trends (Table 3; Figure 1B). Esophageal cancer deaths fell from 22,353 to 13,002, with the age-standardized mortality rate (ASMR) decreasing from 5.15 to 1.79 per 100,000 (EAPC −4.08%). Stomach cancer–related deaths declined from 54,445 to 30,568, and ASMR dropped from 10.05 to 3.66 per 100,000 (EAPC −3.50%). Liver cancer deaths remained relatively stable (27,360 in 1990 vs. 28,127 in 2021), while ASMR decreased from 4.96 to 3.38 per 100,000 (EAPC −1.58%). Colon and rectum cancer deaths rose modestly from 22,619 to 24,476, yet ASMR fell from 4.00 to 3.00 per 100,000 (EAPC −1.13%). Gallbladder and biliary tract cancer deaths increased from 1731 to 1946, even as ASMR declined from 0.38 to 0.28 per 100,000 (EAPC −1.01%). Pancreatic cancer deaths rose from 5687 to 8887, with ASMR remaining essentially unchanged (1.05 vs. 1.06 per 100,000; EAPC −0.04%).

Similarly, the AAPC analysis revealed distinct patterns as well (Table 2). For esophageal cancer, the mortality AAPC was −1.750 (95% CI: −2.000 ~ −1.500, *p* < 0.001), and for stomach cancer, it was −1.818 (95% CI: −1.983 ~ −1.653, *p* < 0.001), reflecting a continuous decline in their mortalities. The mortality AAPC of liver cancer was 0.060 (95% CI: −0.210 ~ 0.330, *p* = 0.663), showing no statistically significant change. Colorectal cancer had a mortality AAPC of 0.164 (95% CI: 0.003 ~ 0.326, *p* = 0.046). Gallbladder and biliary tract cancer presented a mortality AAPC of 0.370 (95% CI: 0.224 ~ 0.516, *p* < 0.001), and pancreatic cancer had a mortality AAPC of 1.463 (95% CI: 1.367 ~ 1.560, *p* < 0.001). These findings indicated that the mortality trends varied with the increase in incidence for different cancer types.

Overall, traditional upper-GI malignancies such as stomach and esophageal cancers have exhibited pronounced declines in both incidence and mortality over the past three decades, while liver cancer has followed a more gradual downward trend since 2000. In contrast, colorectal and biliary tract cancers have steadily risen in incidence, with colorectal cancer becoming the most prevalent GI malignancy by the 2010s, although their mortality rates have largely plateaued or edged lower. Pancreatic and gallbladder/biliary tract cancers have shown only modest shifts, with slight increases in incidence alongside largely stable mortality for pancreatic cancer and gradual incidence rises paired with small declines in biliary tract cancer death rates (Figure 1).

### 3.2. Subtype Distribution of Early-Onset Gastrointestinal Cancer Age Groups

Incidence of early-onset gastrointestinal malignancies increased progressively from adolescence into middle age and peaked in the 45-to-49-year age group. In every cohort, colon and rectum cancer accounted for the largest proportion of cases, representing more than half of incidents in the youngest group and declining gradually to about one third by age 45 to 49 years. Stomach cancer remained the second most frequent diagnosis across all ages, comprising roughly one quarter to one third of cases. At the same time, the relative shares of liver and esophageal cancers rose with advancing age, while gallbladder and pancreatic cancers remained comparatively rare (Figure 2A).

Gastrointestinal cancer mortality rose steadily with age, increasing from less than one death per 100,000 population in the 15-to-19-year group to nearly fifty per 100,000 in the 45-to-49-year cohort. In every age band, stomach and colorectal cancers together accounted for over half of all deaths, with colorectal cancer responsible for about forty percent of mortality in adolescents before declining to roughly twenty-five percent by middle adulthood, while stomach cancer remained at approximately one quarter of deaths throughout. Liver cancer mortality grew in both absolute numbers and relative share through early adulthood, stabilizing at around one quarter of deaths in the oldest groups. Meanwhile, esophageal and pancreatic cancers each increased their contribution to overall mortality from only a few percent in the youngest cohort to low double figures by age 45 to 49 years. Gallbladder and biliary tract cancers remained a consistently minor cause of death across all ages (Figure 2B).

### 3.3. Age- and Sex-Specific Trends in Early-Onset Gastrointestinal Cancer Incidence and Mortality

Age-specific incidence curves (Figure 3A–F) and corresponding mortality trends (Figure 4A–F) demonstrate a consistent increase in early-onset gastrointestinal cancer risk. Pancreatic cancer remains uncommon until the early thirties before accelerating in both incidence and deaths, particularly among males. Colon and rectum cancer exhibit the most pronounced age gradient, climbing steadily from young adulthood to mid-life and surpassing other sites in burden, especially in men. Stomach and liver cancers increase at intermediate rates across the age span, again more markedly in males, while gallbladder and biliary tract cancers display more gradual, lower-magnitude age-related rises. Esophageal cancer rates climb sharply after the mid-thirties, with a widening sex gap in older age groups. Together, these patterns underscore that early-onset gastrointestinal cancer risk intensifies with age for all sites and remains consistently higher in males throughout early to mid-adult life.

### 3.4. Projected Trends in Early-Onset Gastrointestinal Cancer Incidence and Mortality

We report age-standardized incidence (Figure 5) and mortality (Figure 6) trends from 1990 to 2040, incorporating observed data through 2021 and projected estimates for the subsequent years. Between 1990 and 2021, early-onset stomach, liver, and esophageal cancers experienced steady declines in both age-standardized incidence and mortality. From 2021 onward, BAPC projections indicate these downward trends will continue through 2040. Colorectal and gallbladder-biliary tract cancers showed rising incidence up to 2021 and are predicted to climb further through 2040. Their mortality rates stabilized between 2005 and 2021 and are projected to resume a gradual downward trajectory over the following two decades. Pancreatic cancer incidence and mortality exhibited a modest decline in the early 2000s, followed by a sustained increase through 2021. The projected ASIR of pancreatic cancer is expected to reach its peak around 2030 at 1.30 per 100,000 (95% UI: 1.15–1.45), after which it is predicted to stabilize or slightly decline. Posterior predictive checks indicated good model fit, with coverage probabilities close to nominal levels and no major residual patterns. Posterior summaries of the hyperparameters are provided in Appendix A.

In addition to the BAPC model, we applied the Autoregressive Integrated Moving Average (ARIMA) projection model (Appendix A). The ARIMA forecasts for incidence and mortality demonstrated generally consistent trends with those from the BAPC model, showing increasing incidence for colorectal and biliary tract cancers, and declining trends for stomach and liver cancers.

## 4. Discussion

In this comprehensive analysis of early-onset gastrointestinal cancers in China from 1990 to 2021, we observed divergent temporal patterns across cancer subtypes. Traditional upper-GI malignancies (esophageal, stomach, and liver cancers) exhibited substantial declines in both age-standardized incidence and mortality rates, despite rising absolute case counts for esophageal and liver cancers. In contrast, colorectal and gallbladder/biliary tract cancers demonstrated steady increases in incidence, with colorectal cancer emerging as the predominant early-onset GI tumor by the 2010s. Pancreatic cancer showed a modest upward trend in incidence alongside largely stable mortality. Our projections suggest that these divergent trends will persist through 2040, with continued declines in upper-GI cancers but further rises in colorectal and biliary tract cancers, and a peak of pancreatic cancer ASIR around 2030.

The observed declines in early-onset esophageal and stomach cancers align with nationwide improvements in food preservation [12,13], reductions in Helicobacter pylori prevalence [14,15], and broader implementation of endoscopic screening programs [16]. The modest decrease in liver cancer mortality likely reflects enhanced hepatitis B vaccination coverage and antiviral therapies [17,18,19]. These measures have reduced the incidence of chronic hepatitis B, which is a major risk factor for liver cancer. Additionally, improvements in early detection and broader use of antiviral treatments have helped reduce viral replication and liver damage, contributing to lower mortality rates. By contrast, the rising incidence of early-onset colorectal cancer echoes patterns reported internationally and underscores shifting lifestyle factors, such as dietary Westernization, sedentary behavior, and obesity, that disproportionately affect younger cohorts [20,21]. In addition, increasing prevalence of metabolic syndrome, higher consumption of ultra-processed foods, and alterations in the gut microbiome have been implicated as important contributors to carcinogenesis in early-onset GI cancers. Genetic susceptibility and earlier exposure to carcinogenic dietary and environmental factors may further accelerate this trend, highlighting the multifactorial nature of the rising burden of early-onset colorectal cancer [22]. The increasing burden of gallbladder and biliary tract cancers may relate to rising gallstone prevalence and metabolic comorbidities [23], although detailed data in younger populations remain limited [24,25]. Beyond international explanations, several China-specific factors may also account for the observed trends. National cancer control initiatives, such as large-scale screening programs for upper gastrointestinal and colorectal cancers in China, have contributed to the earlier detection and declining burden of certain cancers.

Several interrelated factors may underlie these subtype-specific trends. For colorectal cancer, early exposure to diets high in red and processed meats, low fiber intake, and components of metabolic syndrome are well-established drivers [26]. A meta-analysis found that a high intake of ultra-processed foods was associated with a 30% increased risk of colorectal cancer [27]. Genetic predispositions and alterations in the gut microbiome may also accelerate carcinogenesis in younger individuals [27,28]. The uptick in biliary tract cancers may reflect increases in cholelithiasis associated with obesity and dyslipidemia, alongside environmental exposures in certain regions [29,30]. Pancreatic cancer’s projected peak around 2030 could be linked to persistent smoking rates, chronic pancreatitis, and rising diabetes prevalence in patients aged 15–49 years [31,32]. Our projections further indicate that the age-standardized incidence rate of pancreatic cancer will likely peak around 2030 at 1.30 per 100,000 (95% UI: 1.15–1.45), which could be explained by several plausible factors. First, reductions in smoking prevalence in younger cohorts could contribute to lowering future incidence, given that smoking is a major established risk factor for pancreatic cancer. Second, improvements in diabetes management and metabolic health may reduce related risks over time. Third, advances in early detection strategies, including biomarkers and imaging technologies, together with emerging therapeutic options, could improve survival and gradually lessen disease burden. While these determinants are not explicitly modeled in the BAPC framework, they provide reasonable explanations consistent with current epidemiological evidence.

These findings carry important implications for cancer control in China. First, the sustained declines in upper-GI cancers attest to the success of the existing prevention strategies, including continued investment in Helicobacter pylori eradication, aflatoxin reduction, and viral hepatitis control remains essential [33]. Second, the burgeoning burden of early-onset colorectal cancer argues for reconsideration of screening guidelines: lowering the starting age for colonoscopy or integrating non-invasive stool-based tests for individuals under 50 could facilitate earlier detection [2]. In addition, integration of stool-based non-invasive screening methods, such as fecal immunochemical tests (FITs) or fecal DNA assays, may provide cost-effective and scalable alternatives, particularly in resource-limited settings. Third, rising gallbladder and biliary tract cancers highlight the need for risk stratification among patients with gallstones or metabolic syndrome, potentially via ultrasonographic surveillance. Tailored surveillance strategies targeting such high-risk groups may further enhance early detection and reduce disease burden. Finally, the projected peak in pancreatic cancer emphasizes the urgency of research into early biomarkers and novel risk-reduction strategies targeting modifiable exposures in early-onset populations [34].

Broader contextual factors may also help explain the divergent cancer trends observed. Rapid economic development and urbanization in China have contributed to lifestyle transitions, including dietary shifts toward higher fat and processed foods, and reduced physical activity. Environmental influences, such as air and water pollution, may also play a role in shaping cancer risks [35,36]. At the same time, demographic transitions, including population aging and changing reproductive patterns, are likely to affect cancer incidence in complex ways [37]. These qualitative considerations provide important context for interpreting national cancer trends and underscore the multifactorial nature of early-onset gastrointestinal cancers.

Our findings are generally consistent with the most recent national cancer registry report from the National Cancer Center of China [38], which reported an estimated 4.82 million new cancer cases and 3.21 million cancer deaths in 2022. Specifically, that report similarly documented declining age-standardized incidence and mortality rates for esophageal, stomach, and liver cancers, while colorectal cancer incidence continued to rise to 0.56 million new cases, making it the second most common cancer in China. Notably, pancreatic cancer was one of the few major cancers with both incidence and mortality still increasing, underscoring its emerging public health challenge.

This study provides the first comprehensive, 31-year analysis of early-onset gastrointestinal cancers in China, with projections to 2040 using a Bayesian age–period–cohort model. Compared to prior global or regional analyses, our work offers China-specific insights by linking observed trends to national screening initiatives, dietary transitions, and healthcare disparities, offering a comprehensive picture of early-onset GI cancer burden. However, this study has several important limitations. First, the disparities in healthcare access between urban and rural areas in China may influence the reported cancer trends. Limited healthcare access in rural areas can lead to delays in diagnosis, underreporting of cases, and unequal distribution of healthcare resources, which could distort the true burden of cancer in these regions. These factors should be considered when interpreting cancer incidence and mortality data, as they may affect the generalizability of findings from national databases like GBD 2021. Future research should aim to address these disparities through improved data collection in underserved areas. Second, the lack of detailed staging and histological subtype data prevented us from examining trends by tumor stage or pathology. Also, our analysis was limited to age-standardized rates without further subgroup stratification (e.g., urban vs. rural, socioeconomic, or geographic disparities) because such detailed data were not available from the GBD database, which may reduce the granularity of our findings. Third, our age–period–cohort projections are based on past trends and do not account for future changes in screening programs, prevention strategies, or therapeutic advances, so forecasted values should be interpreted with caution. A further limitation is that we were unable to directly compare our results with other established Chinese cancer datasets, such as those from the China National Cancer Registry or the Chinese Academy of Medical Sciences Cancer Hospital, because these datasets are not open-access. As such, we could not include them in our analysis. Future work incorporating direct access to these national datasets would allow for a more comprehensive validation of our projections.

## 5. Conclusions

The divergent trends in early-onset gastrointestinal cancers in China, from declining upper-GI malignancies to rising colorectal and biliary tract cancers, underscore the evolving landscape of cancer burden among patients aged 15–49 years. To address these challenges, China must sustain successful prevention programs for esophageal, stomach, and liver cancers, while urgently expanding targeted screening and risk-reduction strategies for colorectal, gallbladder and biliary tract, and pancreatic cancers in early-onset populations. Such efforts will be critical to mitigate the growing public health impact of GI malignancies in younger cohorts through 2040 and beyond.

## Figures and Tables

**Figure 1 cancers-17-02954-f001:**
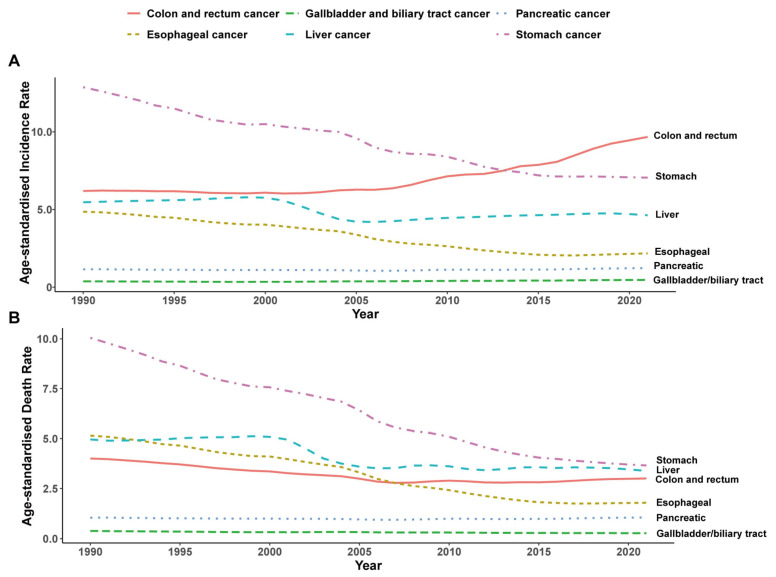
Trends in age-standardized incidence (**A**) and mortality (**B**) rates of early-onset gastrointestinal cancers in China from 1990 to 2021.

**Figure 2 cancers-17-02954-f002:**
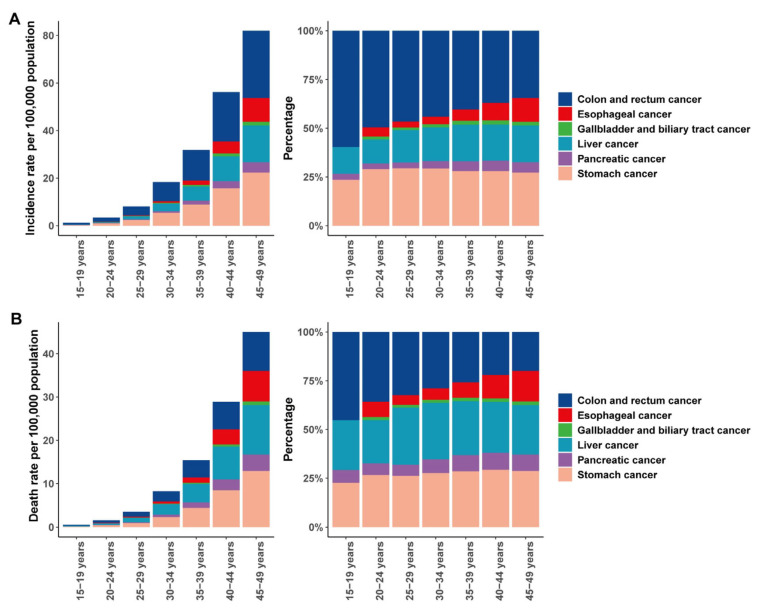
Age-specific incidence (**A**) and mortality (**B**) rates and proportional distributions of early-onset gastrointestinal cancers in China by 5-year age groups.

**Figure 3 cancers-17-02954-f003:**
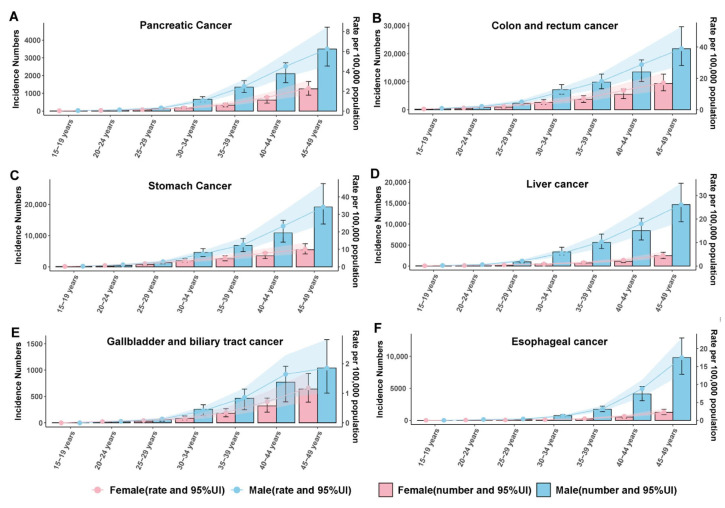
Age- and sex-specific incidence numbers and rates of early-onset gastrointestinal cancers in China for (**A**) pancreatic cancer, (**B**) colon and rectum cancer, (**C**) stomach cancer, (**D**) liver cancer, (**E**) gallbladder and biliary tract cancer, and (**F**) esophageal cancer.

**Figure 4 cancers-17-02954-f004:**
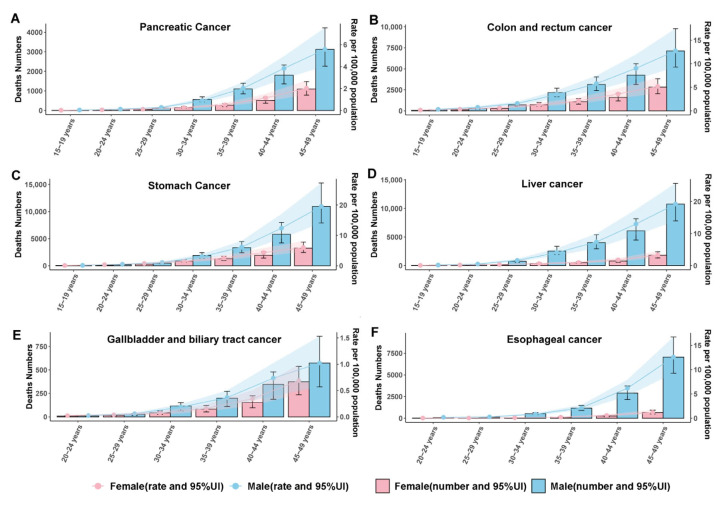
Age- and sex-specific mortality numbers and rates of early-onset gastrointestinal cancers in China for (**A**) pancreatic cancer, (**B**) colon and rectum cancer, (**C**) stomach cancer, (**D**) liver cancer, (**E**) gallbladder and biliary tract cancer, and (**F**) esophageal cancer.

**Figure 5 cancers-17-02954-f005:**
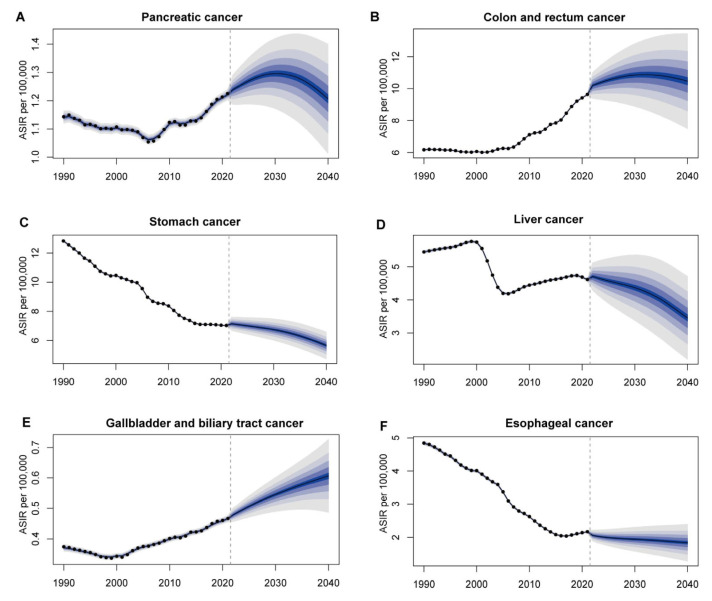
Observed and projected age-standardized incidence rates (ASIRs) of early-onset gastrointestinal cancers in China from 1990 to 2040 for (**A**) pancreatic cancer, (**B**) colon and rectum cancer, (**C**) stomach cancer, (**D**) liver cancer, (**E**) gallbladder and biliary tract cancer, and (**F**) esophageal cancer.

**Figure 6 cancers-17-02954-f006:**
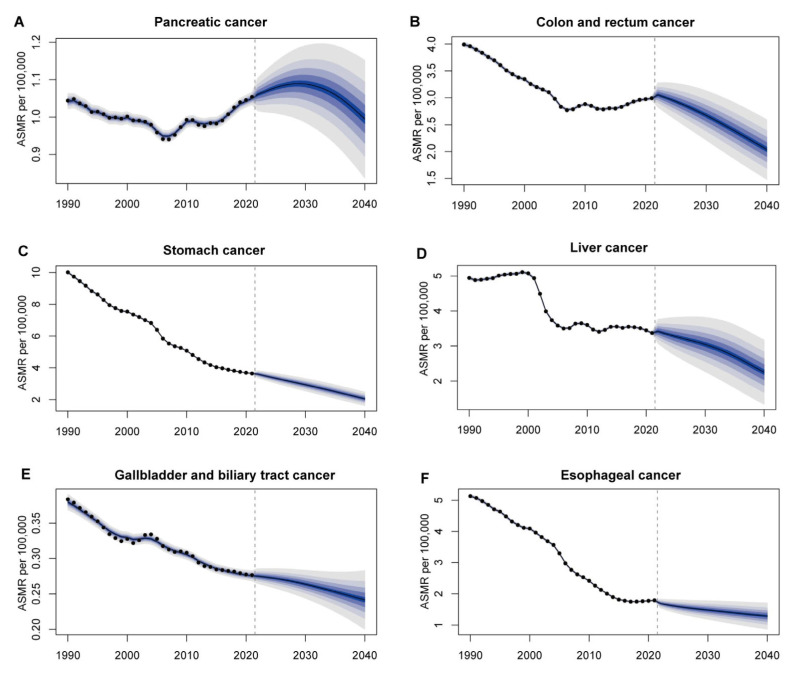
Observed and projected age-standardized mortality rates (ASMRs) of early-onset gastrointestinal cancers in China from 1990 to 2040 for (**A**) pancreatic cancer, (**B**) colon and rectum cancer, (**C**) stomach cancer, (**D**) liver cancer, (**E**) gallbladder and biliary tract cancer, and (**F**) esophageal cancer.

**Table 1 cancers-17-02954-t001:** Comparison of incident cases, age-standardized incidence rates, and estimated annual percent change in gastrointestinal cancers in 1990 vs. 2021. Values are shown as estimates (95% UI) for cases and ASIRs, and estimates (95% CI) for EAPC.

	Number of Cases, 1990	Age-Standardized Rate per 100,000 Population, 1990	Number of Cases, 2021	Age-Standardized Rate per 100,000 Population, 2021	Estimated Annual Percent Change
Esophageal cancer	25,380.20 (21,029.05, 30,165.06)	4.86 (3.98, 5.84)	18,898.87 (14,639.67, 23,588.69)	2.18 (1.69, 2.76)	−3.23 (−3.47, −2.99)
Stomach cancer	70,248.17 (55,605.96, 82,397.39)	12.87 (10.14, 15.20)	58,119.32 (45,912.46, 74,437.82)	7.05 (5.46, 9.01)	−2.10 (−2.21, −2.00)
Liver cancer	30,185.39 (24,838.63, 36,227.37)	5.47 (4.42, 6.67)	38,508.72 (30,544.44, 50,232.43)	4.63 (3.53, 6.04)	−0.84 (−1.14, −0.53)
Colon and rectum cancer	35,027.26 (29,409.35, 40,729.02)	6.19 (5.12, 7.26)	78,692.27 (62,703.11, 96,467.59)	9.67 (7.67, 11.95)	1.44 (1.16, 1.73)
Gallbladder and biliary tract cancer	2034.69 (1359.13, 2459.96)	0.38 (0.25, 0.46)	3893.24 (2417.46, 5088.76)	0.47 (0.29, 0.62)	0.92 (0.73, 1.11)
Pancreatic cancer	6240.82 (5256.33, 7386.02)	1.15 (0.95, 1.37)	10,301.58 (7944.13, 12,878.47)	1.23 (0.95, 1.55)	0.17 (0.04, 0.30)

**Table 2 cancers-17-02954-t002:** Annual Average Percent Change (AAPC) in incidence and mortality of early-onset gastrointestinal cancers in China, 1990–2021. Data are given as estimates (95% CI).

	AAPC of Incidence Rate	*p*-Value	AAPC of Death Rate	*p*-Value
Esophageal cancer	−0.996 (95% CI: −1.213 ~ −0.778)	<0.001	−1.750 (95% CI: −2.000 ~ −1.500)	<0.001
Stomach cancer	−0.579 (95% CI: −0.717 ~ −0.440)	<0.001	−1.818 (95% CI: −1.983 ~ −1.653)	<0.001
Liver cancer	0.721 (95% CI: 0.489 ~ 0.954)	<0.001	0.060 (95% CI: −0.210 ~ 0.330)	0.663
Colon and rectum cancer	2.770 (95% CI: 2.650 ~ 2.890)	<0.001	0.164 (95% CI: 0.003 ~ 0.326)	0.046
Gallbladder and biliary tract cancer	2.163 (95% CI: 1.990 ~ 2.335)	<0.001	0.370 (95% CI: 0.224 ~ 0.516)	<0.001
Pancreatic cancer	1.638 (95% CI: 1.545 ~ 1.730)	<0.001	1.463 (95% CI: 1.367 ~ 1.560)	<0.001

**Table 3 cancers-17-02954-t003:** Comparison of death cases, age-standardized death rates, and estimated annual percent change for gastrointestinal cancers in 1990 vs. 2021. Values are shown as estimates (95% UI) for cases and ASRs, and estimates (95% CI) for EAPC.

	Number of Cases, 1990	Age-Standardized Rate per 100,000 Population, 1990	Number of Cases, 2021	Age-Standardized Rate per 100,000 Population, 2021	Estimated Annual Percent Change
Esophageal cancer	22,353.26 (18,486.33, 26,566.59)	5.15 (4.23, 6.20)	13,002.25 (10,029.61, 16,498.22)	1.79 (1.38, 2.30)	−4.08 (−4.35, −3.81)
Stomach cancer	54,445.40 (43,552.93, 63,913.66)	10.05 (7.93, 11.88)	30,568.44 (23,988.19, 39,506.28)	3.66 (2.84, 4.69)	−3.50 (−3.64, −3.35)
Liver cancer	27,360.02 (22,538.10, 32,769.07)	4.96 (4.01, 6.05)	28,127.08 (22,350.59, 36,519.34)	3.38 (2.58, 4.42)	−1.58 (−1.89, −1.28)
Colon and rectum cancer	22,619.46 (18,985.71, 26,252.24)	4.00 (3.30, 4.69)	24,475.98 (19,469.18, 29,983.10)	3.00 (2.37, 3.74)	−1.13 (−1.36, −0.90)
Gallbladder and biliary tract cancer	1730.99 (1159.43, 2089.77)	0.38 (0.26, 0.47)	1946.30 (1260.37, 2533.62)	0.28 (0.18, 0.37)	−1.01 (−1.09, −0.92)
Pancreatic cancer	5687.11 (4791.69, 6710.96)	1.05 (0.87, 1.25)	8887.05 (6861.73, 11,111.82)	1.06 (0.82, 1.33)	−0.04 (−0.16, 0.08)

## Data Availability

The datasets analyzed during the current study are publicly available from the Global Health Data Exchange (GHDx) website, which hosts the Global Burden of Disease (GBD) 2021 study data. These can be accessed at https://vizhub.healthdata.org/gbd-results/, accessed on 10 July 2025. All estimates used in the analysis were obtained from this repository. Additional data files with extracted results are available upon reasonable request from the corresponding author.

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
