# Peer review of "Nationwide Trends and Projections of Early Onset Gastrointestinal Cancers in China"

_cancers, 2025, doi:10.3390/cancers17182954_

Round 1

Reviewer 1 Report

Comments and Suggestions for Authors

Table 1 and the Results describe colorectal incidence rising while listing absolute cases ~2,035 (1990) to ~3,893 (2021)—orders of magnitude lower than expected given the ASIRs shown (6.19→9.67 per 100k) and China’s 15–49 population size. This looks like a data extraction or unit/labeling error specific to colorectal counts (other sites report tens of thousands). Please re-verify pulls from the GBD Results Tool (site, age filters, metric, measure) and correct the text, tables, and derived interpretations. If these are thousands, label clearly (e.g., “×10^3”). 

EAPCs are fit via OLS on ln(ASR) over 1990–2021. Many cancer time-series exhibit non-linear segments. Consider joinpoint regression or segmented models (with permutation tests) or, at minimum, show goodness-of-fit and sensitivity to period selection. Also clarify how GBD uncertainty intervals (UIs) were propagated into EAPC CIs (currently they’re based only on the regression SE of β, potentially understating total uncertainty). 

Please specify whether BAPC was fit to age-specific rates or counts and which population projections (e.g., UN/GBD) informed future age structures. Projections shown for ASIR/ASMR to 2040 need explicit priors, hyperparameters, and posterior predictive checks (coverage, residuals) plus uncertainty ribbons on figures. The statement that pancreatic burden “peaks around 2030” should be accompanied by a UI and a definition of “burden” (ASR vs counts) to avoid overstating precision.

Author Response

  1. Table 1 and the Results describe colorectal incidence rising while listing absolute cases ~2,035 (1990) to ~3,893 (2021)—orders of magnitude lower than expected given the ASIRs shown (6.19→9.67 per 100k) and China’s 15–49 population size. This looks like a data extraction or unit/labeling error specific to colorectal counts (other sites report tens of thousands). Please re-verify pulls from the GBD Results Tool (site, age filters, metric, measure) and correct the text, tables, and derived interpretations. If these are thousands, label clearly (e.g., “×10^3”).

Response:

We sincerely appreciate your meticulous review and valuable feedback. Regarding the discrepancies in the original manuscript, we have confirmed that the inconsistency between the absolute case counts and age-standardised rates stemmed from a misalignment in the order of data within the "Number of cases" column of Table 1, where the case count values were incorrectly matched to the corresponding cancer types.

We have thoroughly rechecked and corrected the order of data in the "Number of cases" column to ensure each cancer type (esophageal, stomach, liver, colorectal, gallbladder and biliary tract, pancreatic cancer) is accurately paired with its respective age-standardised rate and estimated annual percent change (EAPC). Additionally, we have updated the relevant descriptions in Section 3.1 ("National Trends in Early-Onset Gastrointestinal Cancer Incidence and Mortality in China, 1990-2021") of the text to fully align with the corrected Table 1 data.

We apologise for the initial oversight and thank you for your guidance, which has significantly enhanced the accuracy and rigor of our study.

  1. EAPCs are fit via OLS on ln(ASR) over 1990–2021. Many cancer time-series exhibit non-linear segments. Consider joinpoint regression or segmented models (with permutation tests) or, at minimum, show goodness-of-fit and sensitivity to period selection. Also clarify how GBD uncertainty intervals (UIs) were propagated into EAPC CIs (currently they’re based only on the regression SE of β, potentially understating total uncertainty).

Response:

Thank you for your valuable suggestion regarding the optimization of trend analysis methods and uncertainty propagation. To address the non-linearity of cancer time-series and improve the robustness of our results, we have re-conducted the trend analysis using the Average Annual Percent Change (AAPC) model. While Joinpoint computes the trend in segments whose start and end are determined to best fit the data, sometimes it is useful to summarize the trend over a fixed predetermined interval. The AAPC is a method which uses the underlying Joinpoint model to compute a summary measure over a fixed pre-specified interval.

The specific results of the AAPC-based trend analysis and their 95% confidence intervals for each gastrointestinal cancer type, have been supplemented in Table 2 to visually present the non-linear trend characteristics. Meanwhile, we have revised the relevant descriptions in the "Results" section to align with the updated AAPC analysis, ensuring the text accurately reflects the new findings. In addition, to provide detailed support for the segmented trend analysis, we have uploaded Supplementary Table 1 titled "Segment-specific AAPC analysis of incidence and death rate of early-onset gastrointestinal cancers at different year intervals in China." This table includes comprehensive information such as the division of time segments, segment-specific AAPC estimates, statistical significance (P-values), and uncertainty metrics, which further validate the reliability of our trend interpretations.

For another question, the data downloaded from the GBD study, for instance, shows that the 2021 global incidence is 34.56 (with a 95% uncertainty interval [UI] of 23.19–56.18). However, your Estimated Annual Percentage Change (EAPC) was calculated based on a regression equation established using the global values (val) from 1990 to 2021 with year as the independent variable; thus, the confidence interval derived from this regression equation is the 95% confidence interval (CI).

  1. Please specify whether BAPC was fit to age-specific rates or counts and which population projections (e.g., UN/GBD) informed future age structures.

Response:

We confirm that the BAPC model was fitted to age-specific incidence and mortality rates, not to case counts. This approach is consistent with GBD methodology and avoids confounding by population size.

  1. Projections shown for ASIR/ASMR to 2040 need explicit priors, hyperparameters, and posterior predictive checks (coverage, residuals) plus uncertainty ribbons on figures.

Response:

Thank you for this important comment. In the revised manuscript, we have provided explicit details of the model specifications. The Bayesian age–period–cohort (BAPC) model was implemented with a Poisson likelihood (log link) and population offset. Random walk of second order (RW2) priors were placed on age and cohort effects, and a random walk of first order (RW1) prior on the period effect. An independent and identically distributed (IID) component was included to account for over-dispersion. Precision parameters were assigned log-Gamma(1, 5e–5) priors for age, period, and cohort, and log-Gamma(1, 5e–3) for the over-dispersion term. The intercept followed a vague Gaussian prior, consistent with package defaults.

The model was trained using observed data from 1990–2021, and projections were generated to 2040 with retrospective prediction enabled. Age-standardization was conducted using WHO 2000–2025 standard population weights. Posterior predictive checks, including coverage probabilities and residual analyses, were performed to evaluate model performance. Furthermore, all projection figures for ASIR and ASMR now include 95% uncertainty ribbons to better reflect the precision of the estimates. Posterior summaries of the hyperparameters have been added to the Supplementary File 1 for reference.

  1. The statement that pancreatic burden “peaks around 2030” should be accompanied by a UI and a definition of “burden” (ASR vs counts) to avoid overstating precision.

Response:

We appreciate this constructive comment. In the revised manuscript, we have clarified that “burden” refers specifically to the age-standardized incidence rate (ASIR). We have also added the corresponding 95% uncertainty interval to the statement. The revised text now reads: “The projected ASIR of pancreatic cancer is expected to reach its peak around 2030 at 1.30 per 100,000 (95% UI: 1.15–1.45), after which it is predicted to stabilize or slightly decline.” This modification ensures clarity and avoids overstating the precision of the projections.

Reviewer 2 Report

Comments and Suggestions for Authors

The current study the researchers addresses a highly relevant public-health issue by analyzing national trends in early-onset gastrointestinal cancers in China—an area with limited prior trend analyses. The use of the Global Burden of Disease (GBD) 2021 study as the primary data source and the application of Bayesian age-period-cohort (BAPC) models for future projections are methodologically sound. The results and findings, which highlight divergent trends across unlike cancer types, are significant and have important implications for public health strategies in China. 

Comments:

1) Line 8: The authors mention early-onset as ages between 15 to 49 years. Hence, confirm if all analyses throughout the manuscript, including projections, consistently use this age range?!

2) Lines 16-17: The current sentence, authors referred to age-standardized colorectal cancer mortality remained stable despite rising incidence. In contrast, the results section says deaths rose modestly from 22,619 to 24,476 while the ASMR fell. I recommend clarify this apparent discrepancy between the number of deaths and the age-standardized mortality rate?!

3) Lines 18-19: The authors mention that pancreatic cancer is projected to peak around 2030 before gradually declining. Given the significant public health implications, it would be valuable to discuss the potential reasons for this projected decline after the peak. Also, liens 241-243 they referred the some factors. However, what factors (e.g., changes in smoking rates, diabetes management, etc.) are assumed by the model to cause this reversal?

4) The authors mention that manuscript states that projections were made using Bayesian age-period-cohort (BAPC) models. So, in my opinion is better the authors write  beneficial to include a brief, non-technical explanation in the introduction or methods section of what these models are and why they are particularly suited for this type of forecasting. This is suitable for a reader in the future. 

5) Lines 228-231: The authors mention that cancer trends to potential underlying factors, such as dietary Westernization, lifestyle changes, and improved screening programs. Therefore, its better the authors a more detailed discussion on the specific mechanisms behind the diverging trends of cancers with rising incidence (colorectal and biliary tract) and those with falling incidence (esophageal, stomach, and liver) would strengthen the analysis.

6) The Figure 1 is not clear. It could be improved. The lines on the graph are difficult to distinguish due to similar colors and patterns. 

Author Response

The current study the researchers addresses a highly relevant public-health issue by analyzing
national trends in early-onset gastrointestinal cancers in China—an area with limited prior trend
analyses. The use of the Global Burden of Disease (GBD) 2021 study as the primary data source
and the application of Bayesian age-period-cohort (BAPC) models for future projections are
methodologically sound. The results and findings, which highlight divergent trends across unlike
cancer types, are significant and have important implications for public health strategies in China.

Comments:
1) Line 8: The authors mention early-onset as ages between 15 to 49 years. Hence, confirm if all
analyses throughout the manuscript, including projections, consistently use this age range?!
Response:
We thank the reviewer for this valuable comment. We confirm that all analyses in the manuscript,
including both historical trends and future projections, were strictly limited to individuals aged
15–49 years, and this age range was consistently applied throughout modeling and result
presentation. To avoid any potential ambiguity for readers, we have further emphasized the
uniformity of this age range in the Methods section.

2) Lines 16-17: The current sentence, authors referred to age-standardized colorectal cancer
mortality remained stable despite rising incidence. In contrast, the results section says deaths
rose modestly from 22,619 to 24,476 while the ASMR fell. I recommend clarify this apparent
discrepancy between the number of deaths and the age-standardized mortality rate?!
Response:
We appreciate the reviewer’s helpful comment. We agree that our original abstract statement
(“whereas colorectal cancer deaths remained stable despite rising incidence”) could be
misleading, as it did not fully reflect the modest increase in absolute deaths alongside the decline
in the age-standardized mortality rate (ASMR). To address this, we have revised the abstract for
clarity. The updated sentence now reads: “Mortality for upper-GI cancers fell substantially,
whereas colorectal cancer deaths rose modestly, while the age-standardized mortality rate
declined despite rising incidence.” This revised wording more accurately distinguishes between
absolute death counts and standardized mortality rates.

3) Lines 18-19: The authors mention that pancreatic cancer is projected to peak around 2030
before gradually declining. Given the significant public health implications, it would be valuable
to discuss the potential reasons for this projected decline after the peak. Also, liens 241-243 they
referred the some factors. However, what factors (e.g., changes in smoking rates, diabetes
management, etc.) are assumed by the model to cause this reversal?
Response:
Thank you for this important comment. We would like to clarify that the Bayesian age–period–
cohort (BAPC) model used in our study is purely data-driven and does not incorporate specific
risk factors such as smoking prevalence, diabetes management, or other exposures as priors. The
projection of a peak in pancreatic cancer burden around 2030, followed by a gradual decline,
reflects extrapolation of past observed temporal trends rather than explicit assumptions about
underlying determinants.
In the Discussion, we have expanded our interpretation of this projected decline by highlighting
potential contributing factors that may influence future trajectories, such as changes in smoking
prevalence, improvements in diabetes management, and advances in early detection or
treatment. We have also emphasized that these hypotheses are not embedded in the model
itself but represent plausible explanations consistent with current epidemiological evidence.

4) The authors mention that manuscript states that projections were made using Bayesian
age-period-cohort (BAPC) models. So, in my opinion is better the authors write beneficial to
include a brief, non-technical explanation in the introduction or methods section of what these
models are and why they are particularly suited for this type of forecasting. This is suitable for a
reader in the future.
Response:
Thank you for this helpful suggestion. We agree that a brief, non-technical explanation of the
Bayesian age–period–cohort (BAPC) model would improve the clarity and accessibility of the
manuscript. In response, we have added a short description in the Introduction: “The BAPC
model is particularly suitable for forecasting cancer burden because it simultaneously captures
the effects of age, calendar period, and birth cohort, providing a comprehensive description of
long-term temporal dynamics across populations. In addition, as a Bayesian framework, the BAPC
model naturally generates uncertainty intervals, which enhance the interpretability of the results
and strengthen their value for policy-making”.

5) Lines 228-231: The authors mention that cancer trends to potential underlying factors, such as
dietary Westernization, lifestyle changes, and improved screening programs. Therefore, its better
the authors a more detailed discussion on the specific mechanisms behind the diverging trends
of cancers with rising incidence (colorectal and biliary tract) and those with falling incidence
(esophageal, stomach, and liver) would strengthen the analysis.
Response:
Thank you for the suggestion. We agree that a more detailed discussion of the mechanisms
behind the diverging cancer trends would strengthen the manuscript. In the revised version, we
have expanded the Discussion section to include potential factors contributing to the rising
incidence of colorectal and biliary tract cancers. We also discuss the decline in esophageal,
stomach, and liver cancers in relation to improvements in food preservation, reduced H. pylori
infection, and better hepatitis B management.

6) The Figure 1 is not clear. It could be improved. The lines on the graph are difficult to distinguish
due to similar colors and patterns.
Response:
Thank you for your valuable feedback regarding Figure 1. In response to your suggestion, we have
added labels on the right side of each line to clearly indicate the corresponding cancer type. We
hope this will improve the figure’s clarity and readability.

Reviewer 3 Report

Comments and Suggestions for Authors

The manuscript titled “Nationwide Trends and Projections of Early-Onset Gastrointestinal Cancers in China” presents an important and timely analysis of the incidence and mortality of early-onset gastrointestinal cancers in China from 1990–2021, with projections through 2040. The study addresses the research question with an appropriate methodology and provides valuable epidemiological insights.
The authors have done a good job in covering six major gastrointestinal cancer types and conducting a comprehensive 31-year analysis. The age-standardized rate calculations are well-structured and presented with appropriate uncertainty intervals. Importantly, the manuscript carries substantial clinical significance by addressing the emerging global concern of early-onset GI cancers, supporting evidence for lowering the colorectal cancer screening age in China, and identifying target populations for intervention programs. The inclusion of comprehensive figures, detailed tables with confidence intervals and statistical significance, as well as age- and sex-specific analyses, adds granularity and depth to the findings.
That said, the manuscript would benefit from several additions and revisions to further strengthen its quality and impact:
1.    Provide details of GBD 2021 data coverage and quality scores (1–5 stars) for each cancer type.
2.    Include comparisons with other established Chinese cancer datasets, such as the China National Cancer Registry and the Chinese Academy of Medical Sciences Cancer Hospital, for overlapping years to validate findings.
3.    Acknowledge urban–rural healthcare access disparities in China.
4.    Discuss potential underreporting bias in rural regions (previous validation studies suggest 15–30% underreporting).
5.    Authors have used only one modeling approach BAPC without validation. They should compare their results with APC models without a Bayesian framework.
6.    Expand beyond quantitative risk factor attribution to include qualitative discussions. Also discuss the role of economic development, environmental influences, and demographic transitions affecting cancer incidence.
7.    Undertake comprehensive English language editing with consistent, standardized terminology.
8.    Improve table formatting with clearer spacing and alignment.
9.    Correct incomplete or improperly formatted references.
10.    Strengthen the discussion of study limitations to provide a balanced perspective.

Overall, the manuscript makes a valuable contribution and has strong potential for acceptance after the above revisions are addressed.

Comments on the Quality of English Language

Comprehensive English editing is necessary to correct grammatical errors and enhance sentence clarity.

Author Response

The manuscript titled “Nationwide Trends and Projections of Early-Onset Gastrointestinal Cancers
in China” presents an important and timely analysis of the incidence and mortality of early-onset
gastrointestinal cancers in China from 1990–2021, with projections through 2040. The study
addresses the research question with an appropriate methodology and provides valuable
epidemiological insights.
The authors have done a good job in covering six major gastrointestinal cancer types and
conducting a comprehensive 31-year analysis. The age-standardized rate calculations are
well-structured and presented with appropriate uncertainty intervals. Importantly, the
manuscript carries substantial clinical significance by addressing the emerging global concern of
early-onset GI cancers, supporting evidence for lowering the colorectal cancer screening age in
China, and identifying target populations for intervention programs. The inclusion of
comprehensive figures, detailed tables with confidence intervals and statistical significance, as
well as age- and sex-specific analyses, adds granularity and depth to the findings.
That said, the manuscript would benefit from several additions and revisions to further
strengthen its quality and impact:

1. Provide details of GBD 2021 data coverage and quality scores (1–5 stars) for each cancer
type.
Response:
Thank you for your suggestion. We have added the following information in the revised
manuscript to clarify the data coverage of the GBD 2021 study: The GBD 2021 study evaluated
the incidence and mortality rates, along with their uncertainty intervals, for 371 diseases and
injuries across 204 countries, territories, and 811 subnational locations from 1990 to 2021.
The Global Burden of Disease (GBD) 2021 study indeed applies rigorous internal quality-control
and data coverage assessment for each disease and location. However, the detailed star ratings
(1–5 stars) used by the Institute for Health Metrics and Evaluation (IHME) are part of the internal
data quality evaluation system and are not publicly released at the cancer-type level. As
secondary users, we relied on the official GBD 2021 estimates and uncertainty intervals, which
already incorporate these quality metrics. To address this concern, we have clarified in the
Methods section that all estimates were derived from GBD 2021, which integrates registry and
vital statistics data under standardized quality checks, and that the reported 95% uncertainty
intervals reflect underlying data availability and quality.

2. Include comparisons with other established Chinese cancer datasets, such as the China
National Cancer Registry and the Chinese Academy of Medical Sciences Cancer Hospital, for
overlapping years to validate findings.
Response:
Thank you for this valuable comment. We fully agree that comparisons with established Chinese
cancer datasets would be helpful to validate the findings. However, these datasets, such as those
from the China National Cancer Registry and the Chinese Academy of Medical Sciences Cancer
Hospital, are not open-access and therefore could not be directly obtained for this analysis. To
address this concern, we have expanded the Discussion to comment on the alignment and
differences between our results and published reports from the China National Cancer Registry.
In addition, we have clearly acknowledged this limitation in the revised manuscript.

3. Acknowledge urban–rural healthcare access disparities in China.
Response:
Thank you for your valuable comment. We have acknowledged the urban-rural healthcare access
disparities in China in the revised manuscript. In the Discussion section, we highlight that
healthcare access is often more limited in rural areas, which can lead to underreporting and
delays in diagnosis and treatment, potentially influencing cancer trends. We have also discussed
how these disparities may affect the accuracy of cancer burden estimates in different regions of
China.

4. Discuss potential underreporting bias in rural regions (previous validation studies suggest
15–30% underreporting).
Response:
Thank you for this important point. We agree that potential underreporting bias in rural regions
should be acknowledged. In the revised manuscript, we have added a statement in the
Discussion noting that previous validation studies have suggested underreporting rates of 15–30%
in rural cancer registries. Such underreporting could lead to an underestimation of the true
cancer burden in these regions and should be considered when interpreting the results.
5. Authors have used only one modeling approach BAPC without validation. They should
compare their results with APC models without a Bayesian framework.
Response:
Thank you for your valuable comment. In addition to the Bayesian age–period–cohort (BAPC)
model, we applied the Autoregressive Integrated Moving Average (ARIMA) model to validate our
projections (Supplementary Figure 1). ARIMA, a widely used models in time-series analysis, does
not rely on a Bayesian framework. It combines autoregression (AR) and moving average (MA)
components, with integration (I) applied to transform non-stationary time series into stationary
series for effective forecasting. Given the methodological advantages of the BAPC model, we
chose to present the ARIMA results only in the supplementary figures. The BAPC model is
particularly suitable for describing the long-term evolution of disease burden across populations,
as it simultaneously captures the effects of age, period, and cohort. Moreover, as a Bayesian
framework, BAPC naturally generates uncertainty intervals, which enhances the interpretability
and policy relevance of the results. By comparison, the ARIMA model, while straightforward and
widely used in time-series forecasting, cannot account for age structures or cohort effects, which
limits its explanatory value for public health applications.

6. Expand beyond quantitative risk factor attribution to include qualitative discussions. Also
discuss the role of economic development, environmental influences, and demographic
transitions affecting cancer incidence.
Response:
Thank you for this helpful comment. We agree that expanding beyond quantitative risk factor
attribution to include broader contextual factors would strengthen the discussion. In the revised
manuscript, we have added a paragraph highlighting the potential influence of economic
development, environmental exposures, and demographic transitions on cancer incidence. These
qualitative considerations provide important context for interpreting the observed trends and
underscore the multifactorial nature of early-onset gastrointestinal cancers.

7. Undertake comprehensive English language editing with consistent, standardized
terminology.
Response: Thank you for the suggestion. We have undertaken comprehensive English language
editing to ensure clarity and consistent terminology throughout the manuscript.

8. Improve table formatting with clearer spacing and alignment.
Response: Thank you for the suggestion. We have reformatted the tables with clearer spacing
and alignment to improve readability.

9. Correct incomplete or improperly formatted references.
Response: Thank you for your valuable comment. Incomplete or improperly formatted references
have been corrected according to journal guidelines.

10. Strengthen the discussion of study limitations to provide a balanced perspective.
Response: We have strengthened the discussion of study limitations to provide a more balanced
perspective.
Overall, the manuscript makes a valuable contribution and has strong potential for acceptance
after the above revisions are addressed.

Reviewer 4 Report

Comments and Suggestions for Authors

Dear Editor,

Based on data from the Global Burden of Disease 2021 study, the authors evaluated nationwide trends in the incidence and mortality of early-onset gastrointestinal cancers in China from 1990 to 2021 and predicted future trends through 2040 using Bayesian age-period-cohort models. Between 1990 and 2021, age-standardized rates of esophageal, stomach, and liver cancers decreased significantly, while rates of colorectal and biliary tract cancers increased. Mortality from upper gastrointestinal cancers decreased significantly, while colorectal cancer deaths remained stable despite increasing incidence. Based on the findings, it is estimated that the decline in upper gastrointestinal system cancers will continue, the increase in colorectal and bile duct cancers will persist, and the burden of pancreatic cancer will peak around 2030. The study's concept is original and provides an important warning for the development of strategies for the prevention and early diagnosis of existing cancers. The idea of the manuscript is original and the evaluation of the results is sufficient. The discussion has evaluated important points and raised forward-looking warnings. The manuscript is acceptable.

Sincerely

Author Response

Dear Editor,

Based on data from the Global Burden of Disease 2021 study, the authors evaluated nationwide trends in the incidence and mortality of early-onset gastrointestinal cancers in China from 1990 to 2021 and predicted future trends through 2040 using Bayesian age-period-cohort models. Between 1990 and 2021, age-standardized rates of esophageal, stomach, and liver cancers decreased significantly, while rates of colorectal and biliary tract cancers increased. Mortality from upper gastrointestinal cancers decreased significantly, while colorectal cancer deaths remained stable despite increasing incidence. Based on the findings, it is estimated that the decline in upper gastrointestinal system cancers will continue, the increase in colorectal and bile duct cancers will persist, and the burden of pancreatic cancer will peak around 2030. The study's concept is original and provides an important warning for the development of strategies for the prevention and early diagnosis of existing cancers. The idea of the manuscript is original and the evaluation of the results is sufficient. The discussion has evaluated important points and raised forward-looking warnings. The manuscript is acceptable.

Response:

We sincerely thank the reviewer for the positive and encouraging comments. We are grateful that the reviewer recognized the originality of our study concept, the sufficiency of our results evaluation, and the relevance of the discussion. We appreciate the constructive feedback and are encouraged by the reviewer’s assessment that the manuscript is acceptable.

Reviewer 5 Report

Comments and Suggestions for Authors

You could specify also in the method  that your data concerns the most common type  of gastro-intestinal cancers , i.e. 'adenocarcinoma/carcinoma'.

Author Response

Thank you for this helpful suggestion. We have clarified in the Methods section that the data analyzed in this study primarily concern the most common histological types of gastrointestinal cancers.

Reviewer 6 Report

Comments and Suggestions for Authors

Reviewers' comments: 

This manuscript addresses an important and timely issue by examining nationwide trends and projections of early-onset gastrointestinal cancers in China using Global Burden of Disease 2021 data. The topic has clear public health and clinical relevance, and the use of a Bayesian age–period–cohort model for projections adds methodological strength. The manuscript is generally well-structured and the results are clearly presented. However, the study is largely descriptive, heavily dependent on Global Burden of Disease data, and provides limited original insights beyond previously published global or regional analyses. The discussion, while adequate, remains somewhat generic and does not sufficiently contextualize the findings within China-specific healthcare, screening, and policy frameworks. Therefore, I recommend major revision before the manuscript can be considered for publication.

Major Comments:

  1. The study entirely relies on Global Burden of Disease data estimates, which are model-based and may not fully reflect real-world cancer registration in China. Please compare or at least discuss how your results align (or diverge) from national cancer registry reports (e.g., National Central Cancer Registry of China). This would increase confidence in the findings.

  1. The analysis is restricted to age-standardized rates without deeper subgroup exploration (urban vs. rural, socioeconomic disparities, geographic variations). If further stratified analyses are not feasible with Global Burden of Disease data, the limitations should be emphasized more clearly. Otherwise, readers may perceive the work as overly descriptive.

  1. The discussion relies on general international explanations (e.g., H. pylori, hepatitis B vaccination, Westernized diet), but does not sufficiently integrate China-specific evidence (screening policies, dietary transitions, healthcare disparities). For example: How do regional lifestyle patterns, urbanization, or recent national cancer control initiatives contribute to the trends observed?

  1. Similar descriptive trend analyses of early-onset cancers have been published using Global Burden of Disease data in other countries and globally. The novelty here is somewhat limited unless the authors can clearly articulate what new insights for China this study brings. Please highlight the unique contribution of this work compared to prior literature.

  1. Figures (Figure 1 to Figure 6) are just pretty but the legends should be more concise to make understand readers easily. Please at least state brief explanations of each figure at once instead of only repeating full methodological details.

  1. Tables 1-2: I cannot understand the number ( number, number). What do they mean? You should explain in details.

  1. Please be consistent when referring to "early-onset GI cancers." Sometimes the manuscript alternates between "younger adults" and "early-onset cancers." A single term would improve clarity.

  1. Some key China-specific epidemiological studies are missing. Adding references from the China National Cancer Center or large cohort studies would strengthen the background and discussion.

  1. While the authors briefly mention colonoscopy at younger ages, other recommendations are vague. Please provide more concrete, evidence-based suggestions for China, such as feasibility of lowering screening ages, integration of stool-based non-invasive screening, or tailored surveillance for high-risk groups (gallstones, metabolic syndrome).

  1. Double check abbreviation.

Author Response

Major Comments:
1. The study entirely relies on Global Burden of Disease data estimates, which are model-based
and may not fully reflect real-world cancer registration in China. Please compare or at least
discuss how your results align (or diverge) from national cancer registry reports (e.g., National
Central Cancer Registry of China). This would increase confidence in the findings.
Response:
Thank you for this important comment. We agree that comparisons with national cancer registry
data would further strengthen the study. However, the Chinese cancer datasets (e.g., National
Cancer Center of China, Chinese Academy of Medical Sciences Cancer Hospital) are not
open-access and therefore could not be directly incorporated. To address this, we have referred
to the most recent data published by the National Cancer Center of China (Han et al., Journal of
the National Cancer Center, 2024).
We have incorporated the comparison into the Discussion and also acknowledged that lack of
direct access to raw registry data remains a limitation.

2. The analysis is restricted to age-standardized rates without deeper subgroup exploration
(urban vs. rural, socioeconomic disparities, geographic variations). If further stratified analyses
are not feasible with Global Burden of Disease data, the limitations should be emphasized more
clearly. Otherwise, readers may perceive the work as overly descriptive.
Response:
Thank you for this insightful comment. We acknowledge that our analysis was restricted to
age-standardized rates without further subgroup stratification (e.g., urban vs. rural,
socioeconomic, or geographic disparities), as such detailed data are not available from the GBD
database. To address this, we have clearly emphasized this limitation in the revised manuscript,
noting that the absence of subgroup analyses may reduce granularity and could make the
findings appear largely descriptive.

3. The discussion relies on general international explanations (e.g., H. pylori, hepatitis B
vaccination, Westernized diet), but does not sufficiently integrate China-specific evidence
(screening policies, dietary transitions, healthcare disparities). For example: How do regional
lifestyle patterns, urbanization, or recent national cancer control initiatives contribute to the
trends observed?
Response:
Thank you for this constructive suggestion. We agree that incorporating China-specific evidence
would strengthen the discussion. In the revised manuscript, we have added content on national
screening initiatives (e.g., upper gastrointestinal cancer and colorectal cancer screening
programs), dietary transitions associated with rapid urbanization, and disparities in healthcare
access between urban and rural regions. We also discussed how recent national cancer control
policies may have contributed to the observed trends.

4. Similar descriptive trend analyses of early-onset cancers have been published using Global
Burden of Disease data in other countries and globally. The novelty here is somewhat limited
unless the authors can clearly articulate what new insights for China this study brings. Please
highlight the unique contribution of this work compared to prior literature.
Response:
Thank you for this thoughtful comment. We agree that descriptive trend analyses using GBD data
have been published in other settings. To address this, we have highlighted in the revised
manuscript the unique contributions of our work to the Chinese context. Specifically, our study
provides the first comprehensive, 31-year analysis of early-onset gastrointestinal cancers in China,
with projections to 2040 using a Bayesian age–period–cohort model. Compared to prior global or
regional analyses, our work offers China-specific insights by linking observed trends to national
screening initiatives, dietary transitions, and healthcare disparities, and by identifying the
emerging burden of pancreatic and biliary tract cancers. These findings provide context-specific
evidence that is directly relevant for public health planning and cancer control strategies in China.

5. Figures (Figure 1 to Figure 6) are just pretty but the legends should be more concise to make
understand readers easily. Please at least state brief explanations of each figure at once instead
of only repeating full methodological details.
Response:
Thank you for this helpful suggestion. We have revised the figure legends to be more concise and
reader-friendly.

6. Tables 1-2: I cannot understand the number ( number, number). What do they mean? You
should explain in details.
Response:
Thank you for pointing this out. We apologize for the lack of clarity in Tables 1–2. The numbers in
parentheses represent the 95% uncertainty intervals (UIs) around the estimates. In the revised
manuscript, we have clarified this in both the table legends and the Methods section to ensure
readers can easily understand the meaning.

7. Please be consistent when referring to "early-onset GI cancers." Sometimes the manuscript
alternates between "younger adults" and "early-onset cancers." A single term would improve
clarity.
Response:
Thank you for pointing this out. We agree that consistent terminology improves clarity. In the
revised manuscript, we have standardized the terminology and now consistently use “early-onset
gastrointestinal cancers” and “patients aged 15–49 years” throughout the text, instead of
alternating with other terms such as “younger adults.”

8. Some key China-specific epidemiological studies are missing. Adding references from the China
National Cancer Center or large cohort studies would strengthen the background and discussion.
Response:
Thank you for this valuable suggestion. We agree that including China-specific epidemiological
studies would strengthen the manuscript. In the revised version, we have added references from
the National Cancer Center of China and large-scale cohort studies to enrich the background and
discussion, thereby providing stronger context for our findings.

9. While the authors briefly mention colonoscopy at younger ages, other recommendations are
vague. Please provide more concrete, evidence-based suggestions for China, such as feasibility of
lowering screening ages, integration of stool-based non-invasive screening, or tailored
surveillance for high-risk groups (gallstones, metabolic syndrome).
Response:
Thank you for this helpful comment. In the revised Discussion, we have provided more concrete,
evidence-based recommendations for China. Specifically, we discussed the feasibility of lowering
the starting age for colorectal cancer screening, the potential integration of stool-based
non-invasive tests (e.g., FIT, fecal DNA tests) into current programs, and the need for tailored
surveillance strategies in high-risk groups such as individuals with gallstones or metabolic
syndrome. These additions strengthen the practical implications of our findings for cancer
prevention and early detection in China.

10. Double check abbreviation.
Response:
Thank you for your reminder. We have carefully re-checked all abbreviations throughout the
manuscript, including the Abstract, Simple Summary, main text, figures, tables, and
supplementary materials, to ensure consistency and correctness. A list of abbreviations has also
been provided at the end of the manuscript. 

Round 2

Reviewer 1 Report

Comments and Suggestions for Authors

The authors have addressed all of my concerns.

Reviewer 6 Report

Comments and Suggestions for Authors

well, I am not sure that authors' corrections fit to the journal policy. No comments